# Study on the Adsorption Behavior of Polymeric Dispersants to S-ZnF Particles during Grinding Process

**DOI:** 10.3390/ma16031287

**Published:** 2023-02-02

**Authors:** Guanghua Huang, Haohan Wu, Zhijun Liu, Hanlin Hu, Shifeng Guo

**Affiliations:** 1Hoffmann Institute of Advanced Materials, Shenzhen Polytechnic, 7098 Liuxian Blvd, Nanshan District, Shenzhen 518055, China; 2Shenzhen Key Laboratory of Smart Sensing and Intelligent Systems, Shenzhen Institute of Advanced Technology, Chinese Academy of Sciences, 1068 Xueyuan Avenue, Shenzhen University Town, Shenzhen 518055, China; 3Guangdong Provincial Key Lab of Robotics and Intelligent System, Shenzhen Institute of Advanced Technology, Chinese Academy of Sciences, 1068 Xueyuan Avenue, Shenzhen University Town, Shenzhen 518055, China

**Keywords:** water-based dispersants, grinding, S-ZnF particles, adsorption thermodynamic, adsorption kinetics

## Abstract

Three sodium polyacrylate copolymers PD0x (Poly acrylic acid-co-sodium 4-vinylbenzenesulfonate or PD01; Poly acrylic acid-co-sodium 4-vinylbenzenesulfonate-co-hydroxyethyl methacrylate or PD02 and Poly methyl methacrylate-co-acrylic acid-co-sodium 4-vinylbenzenesulfonate-co-hydroxyethyl methacrylate or PD03) were synthesized as water-based dispersants for grinding red–brown pigment ZnFe_1.2_Cr_0.8_O_4_ particles prepared by the solid phase method (S-ZnF). The particle size distribution, viscosity of suspensions, and adsorption capacity of dispersants were explored by laser particle size analysis, viscometer, and thermogravimetry (TG), respectively. The application of 2 wt.% dispersant PD02 in the S-ZnF suspension ground for 90 min can deliver a finer product with the narrower particle size distribution. The added dispersant PD02 in the grinding process of the S-ZnF particles exhibits a suitable viscosity of the suspension and generates more hydrogen bonds on the S-ZnF particle surface. The sulfonic acid groups (SO_3_^−^) and carboxylic acid groups (-COO^−^) in the dispersant PD02 can also provide a strong charge density, which is favorable for the dispersion and grinding of the S-ZnF particles in the suspensions. Furthermore, the adsorption behavior of polymeric dispersant PD02 adsorbed on the S-ZnF particles surface was simulated and analyzed by adsorption thermodynamic models and adsorption kinetic models. It is indicated that the adsorption thermodynamic behavior of dispersant PD02 adsorbed on the S-ZnF particles surface follows the Langmuir model, and the adsorption process is endothermic and a random process with increased confusion during the grinding process. In addition, the adsorption kinetics of dispersant PD02 adsorbed on the S-ZnF particles surface are more in line with the pseudo-first-order kinetic models. Therefore, the adsorption process of dispersant PD02 on the S-ZnF particles surface can be considered as a single-surface adsorption process.

## 1. Introduction

At present, among all kinds of red ceramic pigments, the zirconium-silicate-coated cadmium sulfide selenide (CdS_x_Se_1−x_@ZrSiO_4_) red pigments can maintain a crimson color at high temperatures [1]. However, when the CdS_x_Se_1−x_@ZrSiO_4_ red pigment particles are ground and prepared as ink, the color changes from a strong red to a yellow peach color which cannot meet the requirements of ceramic ink. In addition, the preparation conditions of the CdS_x_Se_1−x_@ZrSiO_4_ red pigments are quite harsh and the raw materials are also toxic, which seriously hinders the large-scale application of the CdS_x_Se_1−x_@ZrSiO_4_ red pigments [2,3]. The ZnFe_2−x_Cr_x_O_4_ red–brown ceramic pigments are commonly used in red–brown ceramic inks due to their excellent high temperature resistance and chemical stability [4]. The red–brown ceramic inks are prepared by grinding the ZnFe_2−x_Cr_x_O_4_ pigment particles to the submicron level with a stirred bead mill and then adding other additives to meet the requirements of inkjet printing technology.

In recent years, due to the technical advantages and characteristics of the grinding and dispersion method, people have widely used the grinding and dispersion method to prepare different kinds of ceramic ink [5,6,7,8,9,10]. For effective grinding, the type and amount of dispersant, the parameters of the grinding process (e.g., grinding machine type, grinding time, grinding machine speed, etc.), the viscosity of the suspension, and the solid content should be taken into account during the grinding process.

Kuscer et al. prepared a water-borne suspension of TiO_2_ particles with a size less than 570 nm by grinding and added appropriate amounts of nonionic amphiphilic compounds and glycerol to the water-borne suspension to adjust its surface tension and viscosity [11]. Liu et al. synthesized water-based polyacrylic acid dispersant containing hydroxyl group and applied it to wet grinding of the CoAl_2_O_4_ particles. The results show that the polymeric dispersant containing hydroxyl groups is more conducive to particle crushing [12]. Huang et al. believe that during the wet grinding of particles, the grinding suspension with a certain viscosity facilitates particle breakage and thus produces products with small size and narrow particle size distribution [13]. He et al. think that suspension viscosity significantly influences the grindability of industrial minerals in wet ultrafine grinding [14]. Generally, it is believed that the viscosity of ground suspensions can be increased by adding polymeric dispersants during the wet grinding of particles. The main reason is that the polymeric dispersant can absorb on the particle surfaces, which limits the movement of the particles to a certain extent so the viscosity of the ground suspension increases.

It is well known that the adsorption of heavy metal ions by chemical adsorbents in wastewater has attracted wide interest of researchers [15,16,17,18,19,20]. Common adsorbent materials include activated carbon, biocharcoal [21], chitosan, graphene oxide, carbon nanotubes, zeolite [22], clay minerals, etc. In order to describe the adsorption process and mechanism of chemical adsorbents for heavy metal ions in wastewater, researchers usually use adsorption isotherm model and adsorption kinetic model to calculate and simulate them. In the current research, adsorption isotherm models, such as the Henry model, Langmuir model, Freundlich model, and Tempkin model, are widely used. The pseudo-first-order kinetics model, pseudo-second-order kinetics model, Elovich model, and Weber–Morris intraparticle diffusion model are widely used in adsorption kinetics model applications.

At present, the adsorption capacity and factors of polymeric dispersants on the particle surface have been reported [23,24,25,26]. However, little has been reported on the adsorption behavior of polymeric dispersants adsorbed on the particle surface in grinding process.

Hence, in this paper, three sodium polyacrylate copolymers, PD01, PD02 and PD03, were synthesized as water-based dispersants and used for the wet grinding of the S-ZnF particles. Laser particle size analysis and a viscometer were used to analyze the grinding effect and the viscosity of ground suspensions under different conditions, respectively, and the amounts of polymeric dispersant adsorbed on the S-ZnF particle surfaces were calculated by thermogravimetry (TG). In addition, the adsorption behavior of polymeric dispersants adsorbed on the S-ZnF particle surface was analyzed by adsorption thermodynamics and an adsorption kinetics model.

## 2. Experimental Section

### 2.1. Materials

Solvents and reagents were purchased and used without any purification unless noted otherwise. They are Acrylic acid (AA, purity: 99%), Methyl methacrylate (MMA, purity: 99%) and Ammonium persulphate (APS, purity: 99%) purchased from Shanghai Macklin Biochemical Co., Ltd., Shanghai, China; 2-Hydroxyethyl methacrylate (HEMA, purity: 98%), Sodium 4-vinylbenzenesulfonate (SVBS, purity: 90%), Sodium hypophosphite (NaH_2_PO_2_, purity: 98%) and Deuterium oxide (D_2_O, purity: 99.9%) purchased from Energy Chemical Co., Ltd., Shanghai, China; Ethyl alcohol absolute (EtOH, purity: 98%) purchased from Guangzhou Chemical Reagent Co., Ltd., Guangzhou, China; Sodium hydroxide (NaOH, purity: 99%) purchased from Tianjin Fuchen Chemical Reagent Co., Ltd., Tianjin, China. The raw material of red–brown pigment ZnFe_1.2_Cr_0.8_O_4_ particles prepared by solid phase method (S-ZnF) is produced by Foshan Oceno Ceramics Co., Ltd., Foshan, China, and the particle size distribution and photos of S-ZnF pigment raw materials are shown in Figure 1.

### 2.2. Synthesis of Polymeric Dispersants

Figure 2 and Table 1 shows the copolymerization process and synthesis conditions of dispersant PD0x, respectively. Taking the synthesis of dispersant PD03 as an example, Methyl methacrylate (5 g, 0.05 mol), Acrylic acid (36 g, 0.5mol), Sodium 4-vinylbenzenesulfonate (51.5 g, 0.25 mol), Hydroxyethyl methacrylate (6.5 g, 0.05 mol), and deionized water (150 mL) were added to a 500 mL flask equipped with reflux condenser tubes and mixed under magnetic stirring. The temperature of the mixture was adjusted to 70 °C in nitrogen protection. The aqueous solutions of Sodium hypophosphite (15 g, 0.125 mol) and Ammonium persulfate (3 g, 0.013 mol) were dripped into the reaction flask, respectively. When the drop was finished, the reaction temperature was maintained at 80 °C until the monomers were completely reacted. After cooling the reaction solution to ambient temperature, it was neutralized with 50% NaOH aqueous solution to a pH value of 7–8. Then, ethyl alcohol absolute was added into the neutralized reaction solution. After stirring with glass rod, sodium polyacrylate dispersant precipitated from the solution. The dispersant solid was collected and cleaned with the hot ethyl alcohol absolute three times. Finally, this dispersant solid was dissolved in deionized water to form an aqueous solution with a solid content of 40%, i.e., the final dispersant PD03. The synthesis process of dispersants PD01 and PD02 is the same as PD03.

### 2.3. Grinding Process of S-ZnF Particles

Aqueous suspensions of S-ZnF particle raw materials at different solid contents (i.e., 25 wt.%; 35 wt.%, and 45 wt.%) were ground in a model WS-0.3 stirred bead mill (Shenzhen Sanxing Feirong Machine Ltd., Shenzhen, China) at 1995 rpm. The grinding media used were yttrium-stabilized zirconia beads with the sizes of 0.3–0.5 mm, and its volume filling rate was 75 vol.%. The dispersants PD0x (PD01, PD02, and PD03) in different amounts were used in grinding of S-ZnF particle raw materials.

### 2.4. Characterizations

The number-average molecular weight (Mn) and polymeric dispersity index (PDI) of the dispersants PD0x were tested by gel permeation chromatography (GPC, Waters Co., Ltd., Milford, CN, USA). The ^1^H-NMR data and elemental analysis results of dispersant PD0x were tested by Avance III HD 400 MHz NMR spectrometer (Bruker Co., Ltd., Karlsruhe, Germany) and vario EL organic elemental analyzer (Elementar Co., Ltd., Hanau, Germany), respectively. The deuterium solvent used was deuterium oxide (D_2_O). The Fourier transform infrared (FTIR) spectra of samples were determined in a model Vertex 70 FT-IR spectrometer (Bruker Co., Ltd., Karlsruhe, Germany). The pH value of the suspension was adjusted with a diluted sodium hydroxide solution. The viscosity of ground suspensions was measured by a model NDJ-5S viscometer (Shanghai Yoyi lab Co., Ltd., Shanghai, China). The particle size distributions of the ground suspensions were analyzed by a model BT-9300S laser diffraction particle size analyzer (Dandong Bettersize Instrument Ltd., Dandong, China).

The adsorption of dispersant PD0x on the S-ZnF particle surface was measured by a model Q500 thermogravimeter (TA Instruments Co., Framingham, MA, USA) in air at a heating rate of 10 °C min^−1^. The specific surface area of the S-ZnF particles was measured by a model Flow Sorb Ⅲ instrument (Micromeritics Instrument Co., Norcross, GA, USA) based on the BET N_2_ adsorption principle. The adsorption density ($C^{S}$) on the S-ZnF particle surface is calculated by
(1)$$C^{S}=\frac{M_{1}}{S_{BET}}$$
where $M_{1}$ is the mass of dispersant PD0x adsorbed on the surface of the S-ZnF particles per unit mass, and $S_{BET}$ is the specific surface area of the S-ZnF particles.

The above method was used to measure and calculate the adsorption density $C^{S}$ of dispersant PD0x adsorbed on the S-ZnF particle surfaces at different temperatures (i.e., 15 °C, 25 °C, 35 °C, and 45 °C). Then, the Langmuir adsorption model (Equation (2)) and Freundich adsorption model (Equation (3)) were used to fit the adsorption behavior of dispersant PD0x on the S-ZnF particle surface [27], as shown in Equations (2) and (3):(2)$$C^{s}=\frac{C_{m}^{s}K_{L}C^{L}}{1+K_{L}C^{L}}\;$$
(3)$$C^{s}=K_{f}{\left(C^{L}\right)}^{1/n}$$
where $C^{s}$ is the adsorption density on the S-ZnF particle surface (mg/m^2^); $C_{m}^{s}$ is the maximum adsorption density on the S-ZnF particle surface (mg/m^2^); $C^{L}$ is the concentration of dispersant PD0x in suspensions; $K_{L}$ is the equilibrium constant of Langmuir adsorption equation; $K_{f}$ and 1/n are the equilibrium constants of the Freundich adsorption equation.

In addition, the enthalpy change value (ΔH) and entropy change value (ΔS) during the adsorption process of dispersant PD0x adsorbed on the S-ZnF particle surface can be determined by the following Equations [28]:(4)$$\Delta G=-RT\ln K_{d}$$
(5)ΔG=ΔH−TΔS
(6)$$\ln K_{d}=\frac{\Delta S}{R}-\frac{\Delta H}{RT}$$
where *R* = 8.314 J/molk; T is the absolute temperature (k); $K_{d}\;\left(K_{d}=C^{s}/C^{L}\right)$ is the distribution coefficient of the dispersant PD0x, i.e., the ratio of the amount of dispersant PD0x on the S-ZnF particle surface to the amount in the solvent when the adsorption of the S-ZnF particle surface reaches equilibrium. The values of ΔH and ΔS in Equation (6) are calculated from the slope and intercept obtained by fitting a 1/T straight line with $\ln K_{d}$.

In order to evaluate the adsorption kinetics of the dispersant PD0x adsorbed on the S-ZnF particle surface, the absorption density $q_{t}$ of the dispersant PD0x adsorbed on the S-ZnF particle surface at different adsorption time (i.e., 30–210 min) was measured and calculated using the above test methods. Then, the $q_{e}$ and $k_{f}$ values were calculated by pseudo-first-order kinetic model (Equation (7)) and pseudo-second-order kinetic model (Equation (8)) [29,30].
(7)$$q_{t}=q_{e}\left(1-e^{-k_{1}t}\right)$$
(8)$$q_{t}=\frac{k_{2}q_{e}^{2}t}{1+k_{2}q_{e}t}$$
where $q_{e}$ (mg/m^2^) and $q_{t}$ (mg/m^2^) are the adsorption density when the S-ZnF particles are in the adsorption equilibrium state and *t* is adsorption time.

The Elovich model is used to describe the chemical adsorption behavior [31]. It can react with the chemical adsorption process between solid and liquid phases. The Elovich model is Equation (9).
(9)$$\frac{dq_{t}}{dt}=\alpha \cdot e^{-\beta q_{t}}$$

Since Equation (9) is difficult to calculate, it is converted into a linear equation such as Equation (10).
(10)$$q_{t}=a+b\ln t$$
where $a=\frac{1}{\beta }\ln\left(\alpha \beta \right)$, $b=\frac{1}{\beta }$, α is the adsorption rate constant, and β is the activation energy constant.

## 3. Results and Discussion

### 3.1. Water-Based Dispersants PD0x

The FTIR spectra of the dispersants PD0x (PD01, PD02, and PD03) are shown in Figure 3. In Figure 3a, the absorption peak at 2970 cm^−1^ demonstrates the existence of a carbon-carbon single bond (-C-C-). The peaks at 1573 cm^−1^ and 1405 cm^−1^ show the presence of carboxylic acid groups (-COO^−^). The peak at 817 cm^−1^ is the characteristic absorption peak of phenyl group. In Figure 3b, the absorption peak at 1714 cm^−1^ shows the presence of ester groups (-CO-O-) in the monomer HEMA. The absorption peaks at 1189 cm^−1^, 1046 cm^−1^, and 630 cm^−1^ could prove the existence of sulfonic acid groups (SO_3_^−^). In Figure 3c, the absorption peak at 3268 cm^−1^ proved the presence of hydroxyl groups (-OH). The absorption peak at 1720 cm^−1^ shows the presence of ester groups (-CO-O-). The peaks at 1573 cm^−1^ and 1405 cm^−1^ were the characteristic absorption peaks of carboxylic acid groups (-COO^−^). The absorption peak at 817 cm^−1^ was the characteristic absorption peak of phenyl group. These results indicate that the dispersant PD0x was obtained.

The ^1^H-NMR spectra of the dispersants PD0x are shown in Figure 4. In Figure 4(1), the peaks at δ = 7.5 ppm and δ = 6.3 ppm are the characteristic chemical shifts of proton hydrogen in the phenyl group. The peaks at δ = 2.2 ppm and δ = 1.5 ppm prove the existence of a polymer backbone. In Figure 4(2), in addition to the chemical shift in Figure 4(1), the peaks at δ = 3.6 ppm and δ = 3.5 ppm are the characteristic chemical shifts of proton hydrogen on the linked ester group (-CO-O-CH_2_-CH_2_-) in HEMA. In Figure 4(3), in addition to the chemical shift in Figure 4(2), the peak at δ = 3.4 ppm is the characteristic chemical shifts of proton hydrogen in ester group (-CO-O-CH_3_). Thus, it can be seen from Figure 4 that the dispersants PD0x are the target products for this study.

Table 2 shows the elemental analysis results of dispersant PD0x. The presence of element N and absence of element C by the initiator APS used in the synthesis process of dispersant PD0x is shown, and there is element C and no element N in dispersant PD0x. The element N in Table 2 can be considered to be due to the residue of initiator APS, while the element C can be considered to be present in the dispersant PD0x. Therefore, the purity of dispersant PD0x can be known by the ratio of N/C. From the results of elemental analysis in Table 2, it can be seen that the purity of dispersants PD0x synthesized by water-based radical polymerization is more than 90%, which meets the requirements for this study.

Figure 5 shows the GPC spectra of dispersants PD0x, and Table 3 shows the number-average molecular weight (Mn), polymeric dispersity index (PDI), and yield of dispersants PD0x. As shown in Figure 5, the GPC spectra of the dispersant PD0x has a single peak, indicating that the dispersant PD0x is a polymer of one component and meets the requirements of being a dispersant. From Table 3, we can see that the Mn of the dispersant PD0x is between 7000 and 9000, and the PDI is between 3 and 5. Previous studies have shown that dispersants have excellent grinding and dispersion properties when the Mn of dispersant is between 5000 and 10,000 [32]. Thus, the dispersants PD0x can be used as dispersants for grinding and dispersion of the S-ZnF particles according to the above test results.

### 3.2. Grinding of the S-ZnF Particles

Figure 6a shows the particle size distribution and median size (D_50_) of the S-ZnF particles with different dispersants (PD01, PD02, and PD03) at 2 wt.% addition. Clearly, compared to dispersants PD01 or PD03, dispersant PD02 gives a fine product (D_50_ = 0.372 μm) with the narrow distribution, thus indicating the more effective grinding effect. This phenomenon is due to the existence of hydroxyl group (-OH) in the dispersant PD02, which easily adsorbs and wets the S-ZnF particle surface [33]. This is beneficial for the S-ZnF particle breakage. In addition, the dispersant PD01 has a higher PDI value than the other two dispersants (PD03 and PD02), which indicates that the molecular chain length of the dispersant PD01 is not uniform enough to cause poor grinding performance [34]. Figure 6b shows the particle size distribution and median size (D_50_) of the S-ZnF particles with different grinding time. Clearly, with the increase in grinding time, the median size (D_50_) of the S-ZnF particle products decreases gradually. When the grinding time is 90 min, the particle size distribution of the S-ZnF particle products is narrower than other grinding times. Figure 6c shows the particle size distribution and median size (D_50_) of the S-ZnF particles at different amounts of dispersant PD02. Clearly, the median size (D_50_) of the S-ZnF particle products is the smallest and the particle size distribution is narrow when the amount of dispersant PD02 is 2 wt.%. However, when the amount of dispersant PD02 added was too much (5 wt.%) or too little (0.5 wt.%), the particle size of the S-ZnF particle products was coarse, and the particle size distribution was wide. Figure 6d shows the particle size distribution and median size (D_50_) of the S-ZnF particles at different solid contents. Clearly, compared to those with a suspension solid content of 25 wt.% or 45 wt.%, the particle size of the S-ZnF particles product was smaller and the particle size distribution was narrower when the solid content of the grinding suspension was 35 wt.%.

The relationship between the grinding effect of S-ZnF particles and the viscosity of S-ZnF particle ground suspensions is also analyzed in this paper. Figure 7a shows the viscosity of the S-ZnF particles ground in the presence of different dispersants (PD01, PD02, and PD03) at 2 wt.% addition; It is shown in Figure 7a that the S-ZnF particle ground suspensions have significantly higher viscosity when dispersant PD02 or PD03 is added compared to dispersant PD01. It was attributed to the presence of hydroxyl groups (-OH) in the dispersants PD02 and PD03, which more easily adsorbed on the surface of the particles, hindering the movement of the S-ZnF particles in ground suspensions and thus increasing the viscosity of the S-ZnF particle ground suspensions. Figure 7b shows the adsorption curves of the S-ZnF particles ground in the presence of different dispersants (PD01, PD02, and PD03). In Figure 7b, in the presence of dispersant PD02 or PD03, the adsorption density of the S-ZnF particle surface is greater than that in the presence of dispersant PD01. This is because the hydroxyl group in the dispersant PD02 or PD03 is more easily adsorbed on the S-ZnF particle surface to generate hydrogen bonds. Therefore, in the grinding process, the proper viscosity of the particle ground suspensions results in a particle product with fine particle size and narrow particle size distribution (see Figure 6a).

Figure 8a shows the viscosity of ground suspensions at different solid contents. In Figure 8a, the viscosity of ground suspensions increased as the solid content of ground suspensions increased. When the solid content of ground suspensions is 35 wt.%, the viscosity of ground suspensions is 13 mPa.s. Under this condition, particle products with small particle size and narrow particle size distributions can be obtained (see Figure 6d). Figure 8b shows the viscosity of ground suspensions with different addition of dispersant PD02. In Figure 8b, when the amount of dispersant PD02 added was 0.5 wt.%, the viscosity of the ground suspension was 45 mPa.s. This is because that the dispersant PD02 is not enough to cover the surface of the S-ZnF particles during the grinding process. This causes aggregation due to direct contact between the S-ZnF particles, resulting in high viscosity and poor dispersion of the ground suspensions. Thereby, aggregation occurs by direct contact between the S-ZnF particles, which makes the ground suspension highly viscous and poorly dispersible. Thus, the grinding effect of the S-ZnF particles is not good (see Figure 6c). In addition, when the amount of dispersant PD02 added was 5 wt.%, the viscosity of the ground suspension was 32 mPa.s. This is attributed to the viscosity of the dispersant PD02 itself.

### 3.3. Adsorption Thermodynamic Analysis

The adsorption thermodynamic behavior of polymeric dispersant adsorbed on the S-ZnF particles surface is shown as the absorption characteristic curves of dispersant PD02 adsorbed on the S-ZnF particles at different temperatures (i.e., 15 °C, 25 °C, 35 °C and 45 °C) in Figure 9. In Figure 9, the adsorption density of dispersant PD02 on the surface of S-ZnF particles becomes larger with increasing temperature. For further insight into the adsorption thermodynamics of dispersant PD02 adsorbed on the S-ZnF particles surface, the data of their adsorption characteristic curves were fitted with Langmuir and Freundich models (see Equations (2) and (3)), and the obtained results are shown in Table 4. As known from the R^2^ values in Table 4, the adsorption thermodynamic behavior of dispersant PD02 adsorbed on the S-ZnF particles surface follows the Langmuir model.

Figure 10 shows the plot of *Ln(K_d_)* versus 1/*T* for dispersant PD02 adsorbed on the surface of the S-ZnF particles. In Figure 10, the values of thermodynamic parameters Δ*H* and Δ*S* were calculated by fitting with equations 4, 5, and 6, and their results are shown in Table 5. Δ*H* = 7 KJ/mol (Δ*H* > 0) indicates that the adsorption process is endothermic [35], and the additional energy is required during grinding of the particles. This energy can be a mechanochemical energy provided by a stirred bead mill. Δ*S =* 31.2 J/molK (Δ*S* > 0) indicates that the adsorption process of the S-ZnF particles surface and dispersant PD02 is a random process with increased confusion [36].

### 3.4. Adsorption Kinetics Analysis

In addition, we investigated the adsorption kinetics of polymeric dispersant PD02 adsorbed on the S-ZnF surface. Figure 11a shows the adsorption characteristic curve of dispersant PD02 on the S-ZnF particles surface at different adsorption times. From Figure 11a, it is known that the dispersant PD02 has a fast adsorption process on the S-ZnF particle surface in the 30–90 min time period and reaches the adsorption equilibrium within 120 min. This is because there are more adsorption sites (mainly some hydroxyl sites) on the S-ZnF particle surface at the initial stage of adsorption, which makes dispersant PD02 more easily adsorbed on the S-ZnF particles surface. However, with the increase in adsorption time, the adsorption sites on the S-ZnF particles surface tend to be saturated. The pseudo-first-order kinetic model, pseudo-second-order kinetic model, and Elovich model (see Equations (7), (8), and (10)) were used to fit and analyze the adsorption characteristic curves of polymeric dispersant PD02 adsorbed on the S-ZnF particle surfaces at different adsorption times. The results are shown in Figure 11b–d. Table 6 shows the parameters of pseudo-first-order kinetic and pseudo-second-order kinetic parameters for dispersant PD02 adsorbed on the S-ZnF particles surface. Table 7 shows the Elovich model parameters for dispersant PD02 adsorbed on the surface of the S-ZnF particles. From Table 6 and Table 7, it can be seen that the adsorption kinetics of polymeric dispersant PD02 adsorbed on the S-ZnF particles surface are more in line with the pseudo-first-order kinetic models. This may be because the pseudo-first-order kinetic models are more suitable for simulating the comprehensive adsorption process of external diffusion, surface adsorption, and internal diffusion, while there is no microporous structure in the interior of the S-ZnF particles. Therefore, the adsorption process of dispersant PD02 on the surface of S-ZnF particles can be considered as a single-surface adsorption process.

### 3.5. Adsorption Diagram

According to DLVO theory, the dispersion of particles in aqueous suspension mainly depends on the electrostatic repulsion energy between particles produced by compressing the double layer when the particles are close to each other [26,37]. In aqueous suspensions, the charged functional groups (e.g., carboxyl, sulfonate) of the dispersant provide more charge density on the surface of the particles, making the electrostatic repulsion between the particles stronger (see Figure 12a). In addition, the anchoring groups (e.g., hydroxyl groups) of the dispersants can be more easily adsorbed on the surface of particles, resulting in spatial steric resistance between particles (see Figure 12b).

Figure 12(c1) shows the FTIR spectrum of the S-ZnF particles. As shown in Figure 12(c1), the peak at 1635 cm^−1^ is the characteristic peak of the O-H vibration of the S-ZnF particles adsorbing water [38]. The peak at 488 cm^−1^ is attributed to the tensile vibration absorption of Fe(Cr)-O on the S-ZnF surfaces. At 600 cm^−1^, it is considered as a Zn-O bond [39]. In Figure 12(c2), the peaks of S-ZnF particles at 600 cm^−1^ and 488 cm^−1^ before grinding move to the peaks at 592 cm^−1^ and 482 cm^−1^ after grinding, respectively. The characteristic peak of the ester group of dispersant PD02 before adsorption moved from 1714 cm^−1^ to 1708 cm^−1^ after adsorption. This is due to the hydrogen bonding between the hydroxyl group (-OH) in the dispersant PD02 and the hydroxyl group (-OH) on the S-ZnF particle surface, which results in the shifting of the peak positions of the metal/oxygen bond and the ester group. Therefore, hydroxyl groups in the dispersant PD02 adsorbed on the S-ZnF particle surface easily wet the S-ZnF particles, contributing to the grinding of the S-ZnF particles. Additionally, sulfonic acid groups (SO_3_^-^) and carboxylic acid groups (-COO^-^) in the dispersant PD02 can provide strong electrostatic repulsion energy to the S-ZnF particle surface. This facilitates the dispersion of S-ZnF particles during grinding, thus making S-ZnF particles favorable for grinding. The possible adsorption diagram of dispersants PD02 adsorbed on the S-ZnF particles surface is shown in Figure 12d.

## 4. Conclusions

Three sodium polyacrylate water-based dispersants, PD01, PD02, and PD03, were prepared by free radical polymerization for wet grinding of the S-ZnF particles in aqueous suspensions.

By adding 2 wt.% dispersant PD02 to the grinding process of the S-ZnF particle, the product with a fine particle size and narrow particle size distribution is obtained. This is due to the fact that the dispersant PD02 has hydroxyl groups and can easily absorb and wet the surface of the particles, and the S-ZnF particle ground suspension maintains proper viscosity. Additionally, the sulfonic acid groups (SO_3_^−^) and carboxylic acid groups (-COO^-^) in the dispersant PD02 can provide a strong charge density, which is favorable for the dispersion and grinding of the S-ZnF particles in the suspensions.

Adsorption thermodynamic models and adsorption kinetic models were used to analyze the adsorption behavior of dispersant PD02 adsorbed on the S-ZnF particle surface. The adsorption thermodynamic behavior of dispersant PD02 adsorbed on the S-ZnF particles surface follows the Langmuir model. The adsorption process is endothermic (Δ*H* = 7 KJ/mol) and is a random process with increased confusion (Δ*S* = 31.2 J/molK) during the grinding process. In addition, the adsorption kinetics of polymeric dispersant PD02 adsorbed on S-ZnF particle surface conforms to pseudo-first-order kinetics model. The adsorption process of the dispersant PD02 on the S-ZnF particle surface can be considered a single-surface adsorption process.

## Figures and Tables

**Figure 1 materials-16-01287-f001:**
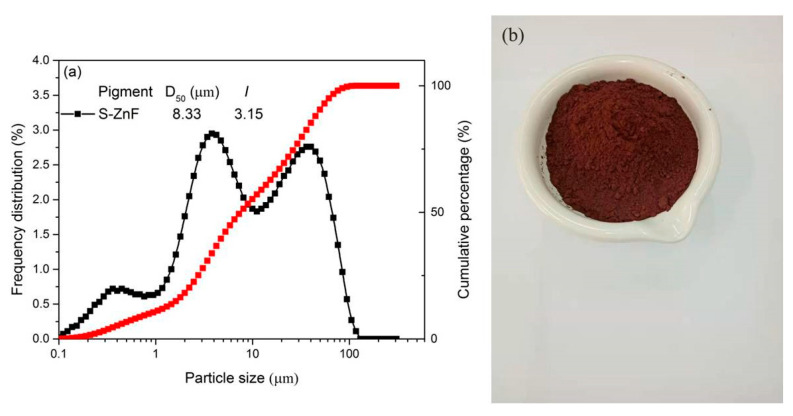
(**a**) The particle size distribution of S-ZnF pigment as a feed sample; (**b**) picture of S-ZnF pigment as a feed sample.

**Figure 2 materials-16-01287-f002:**
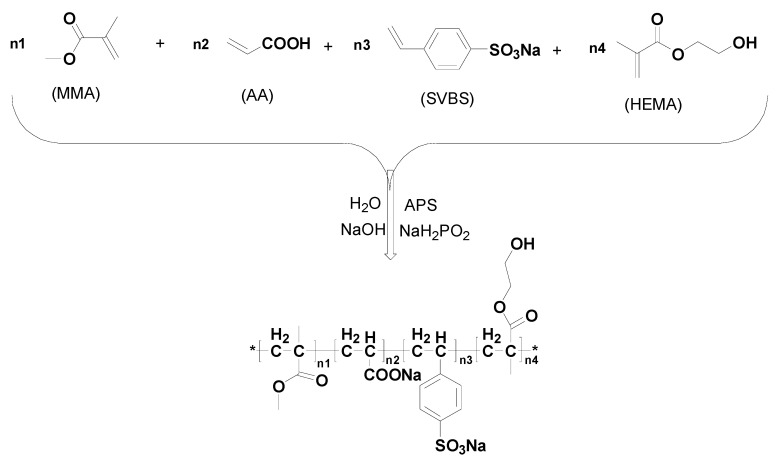
Copolymerization process of dispersant PD0x.

**Figure 3 materials-16-01287-f003:**
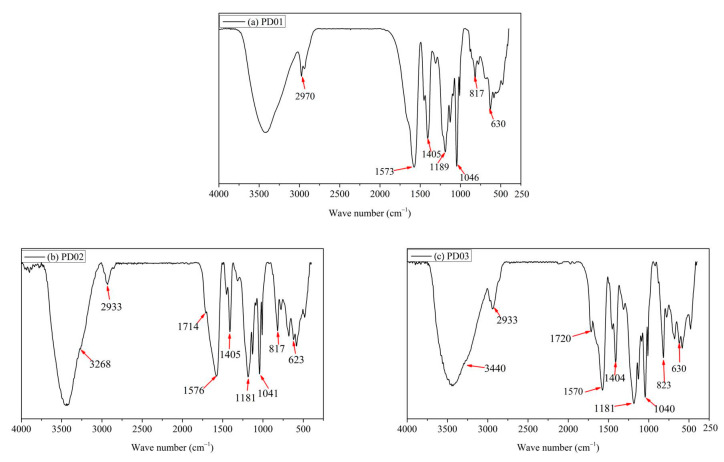
The FTIR spectrum of the dispersants PD0x.

**Figure 4 materials-16-01287-f004:**
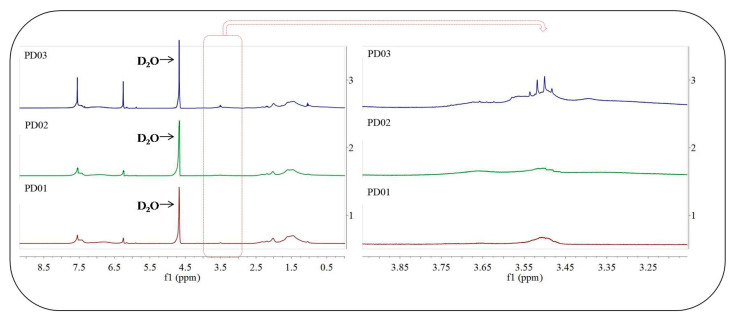
The ^1^H-NMR spectrum of water-based dispersant PD0x.

**Figure 5 materials-16-01287-f005:**
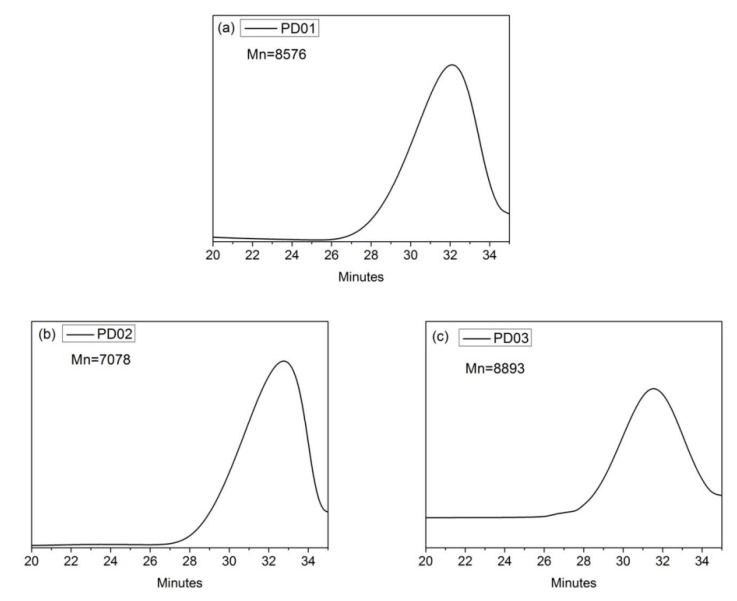
The GPC spectra and Mn of dispersants: (**a**) PD01; (**b**) PD02; (**c**) PD03.

**Figure 6 materials-16-01287-f006:**
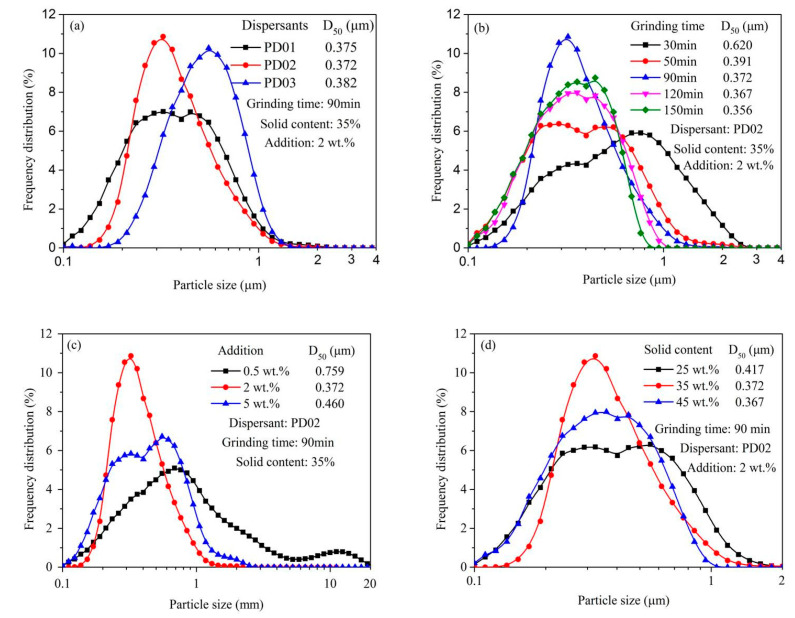
The particle size distribution and median size (D_50_) of the S-ZnF particles with dispersant PD0x (PD01, PD02, and PD03) at 2 wt.% addition (**a**), with different grinding time (**b**), amounts of dispersant PD02 (**c**), and at different solid contents (**d**).

**Figure 7 materials-16-01287-f007:**
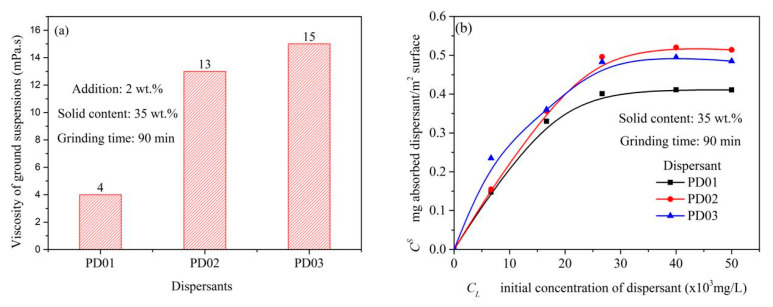
(**a**) Viscosity of the S-ZnF particles ground in the presence of different dispersants (PD01, PD02, and PD03) at 2 wt.% addition; (**b**) adsorption curves of the S-ZnF particles ground in the presence of different dispersants (PD01, PD02, and PD03).

**Figure 8 materials-16-01287-f008:**
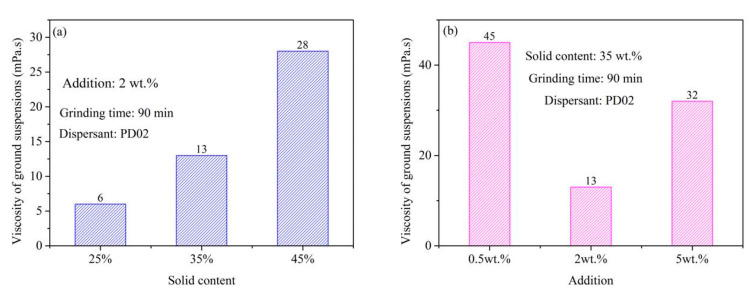
The viscosity of ground suspensions at (**a**) different solid contents; (**b**) with different addition amounts of dispersant PD02.

**Figure 9 materials-16-01287-f009:**
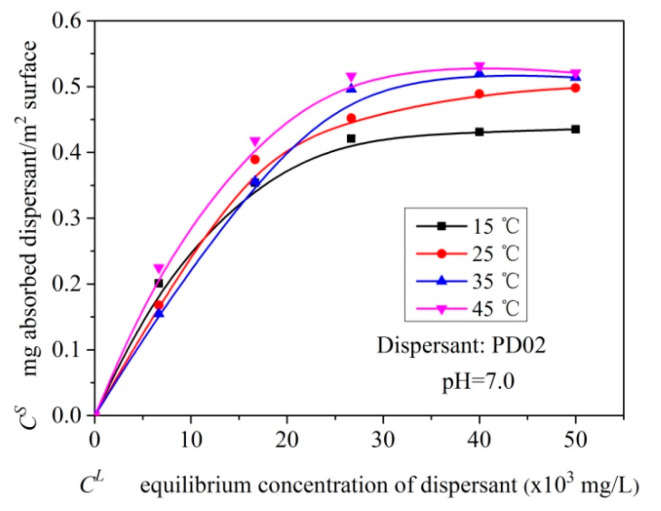
The adsorption isotherms for dispersants PD02 adsorbed on the surface of pigment particles at different temperatures.

**Figure 10 materials-16-01287-f010:**
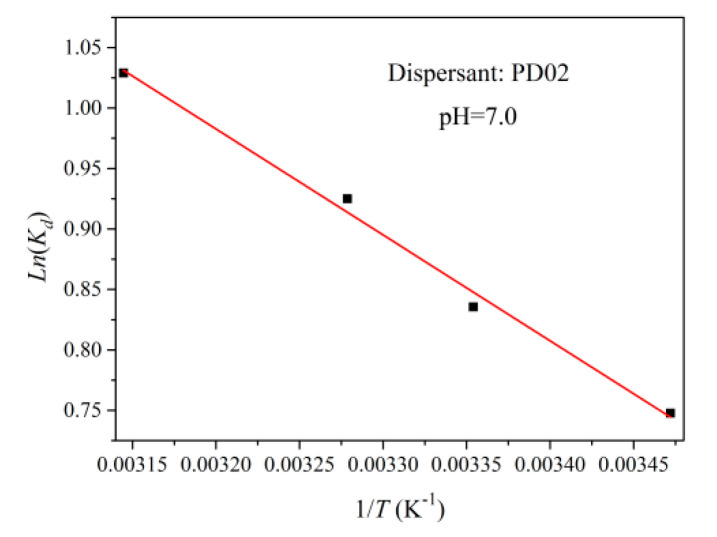
Plot of *Ln*(*K_d_*) versus 1/*T* for dispersant PD02 adsorbed on the surface of the S-ZnF particles.

**Figure 11 materials-16-01287-f011:**
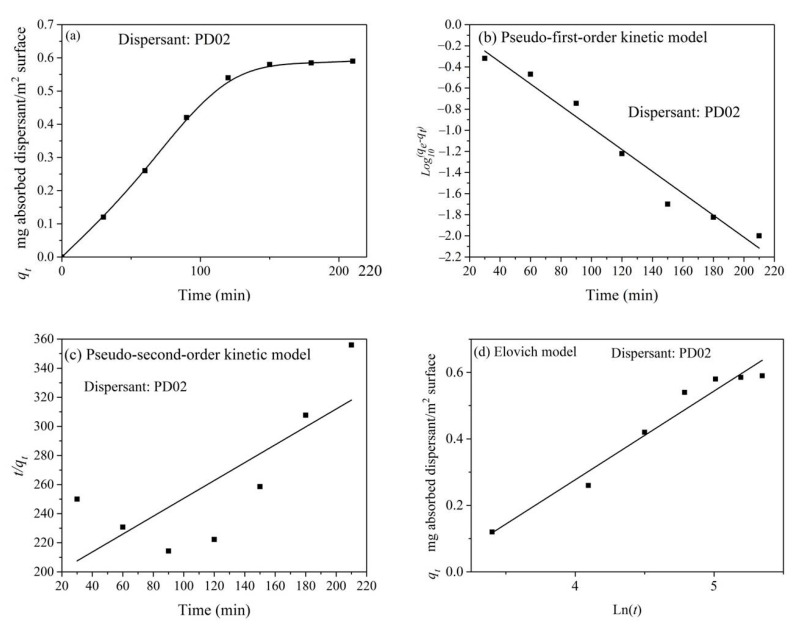
At different adsorption times: (**a**) the adsorption isotherms of dispersants PD02 adsorbed on the S-ZnF particles surface; (**b**) fitting diagram of pseudo-first-order adsorption kinetics model; (**c**) fitting diagram of pseudo-second-order adsorption kinetics model; and (**d**) fitting diagram of Elovich model for dispersants PD02 adsorbed on the surface of the S-ZnF particles.

**Figure 12 materials-16-01287-f012:**
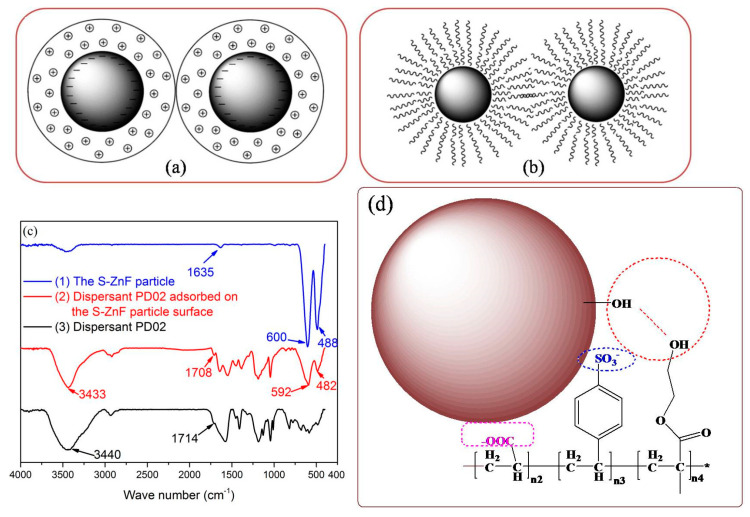
(**a**) Electrostatic repulsion between particles; (**b**) Steric hindrance between particles; (**c**) the FTIR spectra of dispersant PD02 adsorbed on the S-ZnF particle surface; (**d**) The schematics of dispersant PD02 adsorbed on the S-ZnF particle surface.

**Table 1 materials-16-01287-t001:** Synthesis conditions of dispersants PD0x.

PD0x	Mol (n1:n2:n3:n4) ^a^	APS	Mol (APS:NaH_2_PO_2_) ^b^	T/°C	t/h
PD01	0:2:1:0	3 wt.%	1:10	85	10
PD02	0:2:1:0.2	3 wt.%	1:10	75	8
PD03	0.2:2:1:0.2	3 wt.%	1:10	80	8

^a^ Monomers: n1-MMA; n2-AA; n3-SVBS; n4-HEMA. ^b^ Polymerization conditions: APS-3 wt.%; the solid content of reaction fluid −40 wt.%.

**Table 2 materials-16-01287-t002:** Elemental analysis results of dispersant PD0x.

PD0x	N(%)	C(%)	H(%)	S(%)	N/C(%)
PD01	0.09	31.46	3.219	5.072	0.29%
PD02	0.17	31.07	3.483	5.711	0.55%
PD03	0.05	33.70	3.902	3.427	0.15%

**Table 3 materials-16-01287-t003:** Molecular weights and yields of dispersants PD0x.

PD0x	Mn	PDI	Yield
PD01	8576	3.9	92%
PD02	7078	3.5	99%
PD03	8893	4.6	96%

**Table 4 materials-16-01287-t004:** Parameters simulated by the Langmuir equation and Freundich equation.

			Langmuir Equation			Freundich Equation		
T (°C)	Dispersant	Pigment	$K_{L}$ (L/g)	$C_{m}^{s}$ (mg/m^2^)	R^2^	$K_{f}$	1/n	R^2^
15	PD02	S-ZnF	88.1	0.486	0.9489	0.082	0.421	0.9674
25	PD02	S-ZnF	27.6	0.884	0.9652	0.070	0.537	0.8503
35	PD02	S-ZnF	76.2	0.569	0.9901	0.124	0.391	0.9644
45	PD02	S-ZnF	53.5	0.602	0.9781	0.100	0.473	0.9572

**Table 5 materials-16-01287-t005:** Thermodynamic characteristics for dispersant PD02 adsorbed on the surface of the S-ZnF particles.

Dispersant	Pigment	Δ*H* (KJ/mol)	Δ*S* (J/molK)	R^2^
PD02	S-ZnF	7.3	29.75	0.9902

**Table 6 materials-16-01287-t006:** The pseudo-first-order kinetic parameters and pseudo-second-order kinetic parameters for dispersant PD02 adsorbed on the surface of the S-ZnF particles.

		Pseudo-First-Order Kinetic			Pseudo-Second-Order Kinetic		
Dispersant	Pigment	*q_e_* (mg/m^2^)	*k_1_* (min^−1^)	R^2^	*q_e_*(mg/m^2^)	*k_2_* (min^−1^)	R^2^
PD02	S-ZnF	0.987	0.0239	0.9627	1.622	0.0079	0.5177

**Table 7 materials-16-01287-t007:** The Elovich model parameters for dispersant PD02 adsorbed on the surface of the S-ZnF particles.

		Elovich Model				
Dispersant	Pigment	a	b	α	β	R^2^
PD02	S-ZnF	−0.79	0.2667	3.66 × 10^−14^	3.7486	0.9536

## Data Availability

Data is contained within the article.

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
