# Peer review of "Study on the Adsorption Behavior of Polymeric Dispersants to S-ZnF Particles during Grinding Process"

_materials, 2023, doi:10.3390/ma16031287_

Round 1

Reviewer 1 Report

This work focus on the adsorption behaviour of synthetized dispersants to S-ZnF particles during grinding process. The work is interesting and I do not have the major comments to the performed lab work in the field of particle stability, dispersions behaviour etc. However, polymer science part of the manuscript is only superficial. Before accepting the manuscript for the publication also excessive final text correction and proofreading by the authors is needed.
1.    For example, authors use abbreviations PD01-03 in the abstract and highlight that PD02 is the most suitable dispersant, but reader will find the firs information about the structure of the PD0x dispersants only at the page 3 (!) of the manuscript, and only at this page we understand that these dispersants are copolymers. Furthermore, the key fact, that all studied dispersants are sodium polyacrylates is not mentioned in the whole manuscript…(!) This ignoring of the chemistry is not acceptable.
2.    I am not convinced about the author’s term “water-based polymer dispersant”. Like that, one can understand it, that the synthetized polymers are water based. These polymers are not water based, they are polyacrylates. The better wording is needed, e.g. water-based polymeric dispersants, or water-dispersible polyacrylates, or water-soluble polyacrylates for dispersion, or polyacrylates for water based dispersants… Simple google search revel that only your research group uses this “term”. Please can you somehow specify it? I cannot accept this new “term” without clear justification i.e. definition. Water-based inks / resins / glues / paints / dispersants /etc. are OK, but not polymers. 
3.    The scheme of the copolymerisation reaction has to be improved. In the current version, carbon geometry / bond angles are not correct. Please use the clear style as in the organic chemistry text books. I will recommend using clear font (e.g. Arial) for chemical structures and e.g. ACS or Synlett style for bond length etc. It will be much easier to read, and it will look more professional.
4.    With respect to the polymers, their molecular structure and characterization, the manuscript suffers from several deficiencies.  The compound’s characterization is incomplete concerning several important aspects (structure, purity, molar mass). It contains (grave) errors, and is ridden by the misunderstanding of basic concepts.  The attempted structure-property relationship, i.e., identifying the “best” composition of the dispersant, cannot be considered valid on the basis of the data presented. This is not only because only 3 samples were compared (2 points can be always, and 3 points nearly always reasonably well fitted by a straight line, notwithstanding the true correlation function).  In addition, even more importantly, the 3 polymers might differ in a number of other molecular parameters - in addition of the different moieties - that could affect dispersion properties. For example, problem of the copolymerisation parameters is not discussed. Moreover, the authors do not care for a detailed, reliable molecular analysis of the polymers (or are not aware of its necessity), but nevertheless, end up with far-reaching structure-property relationships in their conclusions.
5.    Copolymers are only characterized by FTIR. There are no mass spectra, no 1H or 13C NMR spectra, no elemental analysis (purity=?), SEC.  Furthermore, the paragraph about FTIR is hard to read. I agree with the interpretations but the text must be rewritten. Maybe you can add the table with the key bands to avoid repetitive sentences.
6.    It is not clear how the polymers were isolated after quenching of the copolymerisation reaction.  Dialysis followed by freeze-drying? Please add this information.

Reviewer 2 Report

The present manuscript reports on the “Study on the Adsorption Behavior of Polymer Dispersants to S-ZnF Particles during Grinding Process”. The work is of some interest but seems to be too primitive and lacks in proper scientific support and justification. The synthesized PD and S-ZnF particles is not sufficiently characterized to support the claims. There are many reports exhibiting adsorption process. Thus, in my opinion, the manuscript in its present form cannot be considered for publication. I recommend major revision.

Following are some of the comments/suggestions which will be useful to the authors.

1. First of all, there are many previous works published for adsorption process. The authors seem deliberately avoid those papers. This is unusual, as the authors need to acknowledge the previous literature and compare their work with the similar ones in the literature and demonstrate their research outcomes in terms of advantages and disadvantages. Some of studies are given below need to cited;

 i- Shahid, M., Farooqi, Z. H., Begum, R., Irfan, A., & Azam, M. (2020). Extraction of cobalt ions from aqueous solution by microgels for in-situ fabrication of cobalt nanoparticles to degrade toxic dyes: A two fold-environmental application. Chemical Physics Letters754, 137645.

ii- Arif, M., Farooqi, Z. H., Irfan, A., & Begum, R. (2021). Gold nanoparticles and polymer microgels: Last five years of their happy and successful marriage. Journal of Molecular Liquids336, 116270.

iii- Shahid, M., Farooqi, Z. H., Begum, R., Arif, M., Irfan, A., & Azam, M. (2020). Extraction of cobalt ions from aqueous solution by microgels for in-situ fabrication of cobalt nanoparticles to degrade toxic dyes: A two fold-environmental application. Chemical Physics Letters, 754, 137645.

iv- Arif, M., Shahid, M., Irfan, A., Nisar, J., Wang, X., Batool, N., ... & Begum, R. (2022). Extraction of copper ions from aqueous medium by microgel particles for in-situ fabrication of copper nanoparticles to degrade toxic dyes. Zeitschrift für Physikalische Chemie, 236(9), 1219-1241.

v- Naseem, K., Begum, R., Wu, W., Usman, M., Irfan, A., Al-Sehemi, A. G., & Farooqi, Z. H. (2019). Adsorptive removal of heavy metal ions using polystyrene-poly (N-isopropylmethacrylamide-acrylic acid) core/shell gel particles: adsorption isotherms and kinetic study. Journal of Molecular Liquids, 277, 522-531.

vi- Arif, M. (2022). Complete life of cobalt nanoparticles loaded into cross-linked organic polymers: a review. RSC Advances, 12(24), 15447-15460.

2. Why clearer peak in FTIR was appeared in PD03 at 1720 cm-1 than PD02 and not in PD01. This value is appeared at 1714 cm-1 in PD02. Why?

3. The conditions for adsorption study are missing from each figure such as author should write amount of adsorbent, adsorbate, pH of the medium, rotation time and rotation speed in Figures 7-9.

4. The morphology and size of polymers and after adsorbent are missing from the characterization portion. Characterizations are required for this article.

5. Add more citations in the results and discussion portion to support your results.

6. Re-write you conclusion portion. This portion is the main part of article with reflect the advantage of your results.

7. Indicate the functional groups present in 10 (a,b, and c) pictorially. This point clears the reason of adsorption on adsorbent.

8. The results are not properly explained in discussion portion. Mostly results are given without their proper discussion.

9.  abstract should be improved along with some data obtained during adsorption results.

10. Apply elovich model for chemical kinetics.

11. How can you regenerate the adsorbed material from the surface of adsorbent?

Round 2

Reviewer 1 Report

Dear Authors,

Thank you very much for the revision and update of the manuscript. However, before my recommendation for approval, I will ask you for further elaboration on the analytical data, namely 1H-NMR data, and elemental analysis.

I cannot accept the introduced text in lines (215-220) as it is very superficial.

…The H-NMR spectrums of the dispersants PD0x are shown in Fig. 4. It can be seen from Fig. 4 that the dispersants PD0x are the target products for this study. In addition, Table 2 shows the element analysis results of dispersant PD0x. From Table 2, it can be seen that the purity of dispersants PD0x synthesized by water-based radical polymerization is more than 90%, which meets the requirements of the experiment…

1) Please correct in the whole manuscript, “H-NMR” to “1H-NMR”, specification of hydrogen isotope is obligatory.

 2) Please add the deuterated solved used for the 1H-NMR measurement. It will be beneficial to highlight solvent peaks in the provided spectra (Figure 4) as well.

3) Please describe the 1H-NMR spectra (like what you did for FTIR). It is not the reader’s job to interpret your NMR spectra.

4) The zoomed part of the spectra (right side of Figure 4) must be somehow highlighted (zoom symbol from left spectra to right as it is done e.g., at the presentation of micrographs). It is unclear why you zoomed this part if you do not provide spectra interpretation.

5) Please correct in the whole manuscript, “element analysis” to “elemental analysis”…

6) How do you conclude from elemental analysis that the purity of your product is 90+%? It is not evident to me. Please add data interpretation. The discussion section is obligatory for the MDPI journal Materials.

Reviewer 2 Report

Accept

Author Response

Thank you